# The Antiviral Activities of Poly-ADP-Ribose Polymerases

**DOI:** 10.3390/v13040582

**Published:** 2021-03-30

**Authors:** Mathilde Malgras, Magali Garcia, Clément Jousselin, Charles Bodet, Nicolas Lévêque

**Affiliations:** 1Laboratoire Inflammation Tissus Epithéliaux et Cytokines, Université de Poitiers, 86073 Poitiers, France; mathilde.malgras@univ-poitiers.fr (M.M.); magali.garcia@chu-poitiers.fr (M.G.); clement-jousselin@sfr.fr (C.J.); charles.bodet@univ-poitiers.fr (C.B.); 2Laboratoire de Virologie et Mycobactériologie, CHU de Poitiers, 86021 Poitiers, France

**Keywords:** virus, PARP, antiviral, immunomodulation, viral escape mechanisms

## Abstract

The poly-adenosine diphosphate (ADP)-ribose polymerases (PARPs) are responsible for ADP-ribosylation, a reversible post-translational modification involved in many cellular processes including DNA damage repair, chromatin remodeling, regulation of translation and cell death. In addition to these physiological functions, recent studies have highlighted the role of PARPs in host defenses against viruses, either by direct antiviral activity, targeting certain steps of virus replication cycle, or indirect antiviral activity, via modulation of the innate immune response. This review focuses on the antiviral activity of PARPs, as well as strategies developed by viruses to escape their action.

## 1. Introduction

Poly-adenosine diphosphate (ADP)-ribose polymerases (PARPs) are a family of enzymes responsible for ADP-ribosylation, a reversible and transient post-translational modification of various target proteins including histones, enzymes, transcription factors and even PARPs themselves [1]. PARPs catalyze the transfer of one (mono-ADP-ribosylation or MARylation) or more (poly-ADP-ribosylation or PARylation) ADP-ribose group(s) onto their target proteins using nicotinamide adenine dinucleotide (NAD^+^) as a substrate. ADP-ribosylation can drastically affect the functions of target proteins by modulating their enzymatic activity and by facilitating their ubiquitination, leading to their degradation [2]. PARPs can be found in cells from humans to bacteria and possess a highly conserved C-terminal catalytic domain. Among prokaryotes, many virulence factors of pathogenic bacteria such as diphtheria, cholera and clostridial toxins possess mono-ADP-ribose polymerase activity, causing important dysregulations of host cellular processes, which can lead to cell death [3]. In eukaryotes, PARPs have been identified in, at least, 77 species across five of the six eukaryotic supergroups involved in many cellular activities, such as DNA repair or apoptosis [4]. The human genome encodes 17 PARPs, all sharing a highly conserved sequence in their catalytic domain called the “PARP signature motif”, a characteristic secondary structure that binds NAD^+^. ADP-ribosylation of the target occurs on glutamate, aspartate, cysteine, arginine, serine and lysine residues [4,5,6]. Only five of the 17 human PARPs (PARP1, -2, -3, -5a and -5b) display PARP activity and can promote PARylation. However, most of the human PARPs (PARP6 to PARP12, PARP14, -15 and -16) lack a residue necessary to elongate the ADP-ribose chain, and therefore add a single ADP-ribose to the target, a process called MARylation [7]. Lastly, PARP13 is the only family member with an inactive PARP catalytic domain (Table 1).

PARPs are divided into four subfamilies based on structural domains within the protein outside of the PARP catalytic site. PARP1, -2 and -3 belong to the DNA-dependent PARP subfamily. PARP7, -12 and -13 contain CCCH zinc-finger motifs able to bind RNA. PARP5a and -5b, also known as tankyrases, possess protein-binding ankyrin repeats. Macro-PARPs, including PARP9, -14 and -15, contain two to three macrodomains, which can bind ADP-ribose or its derivatives. Finally, PARP4, -6, -8, -10, -11 and -16 remain unclassified due to the lack of characteristic domain other than the PARP signature [8].

Due to their distinct functional domains, PARPs can play various roles in the cell. PARPs act as transcription regulators through ADP-ribosylation of histones. Since ADP-ribose is negatively charged, PARylation or MARylation of histones leads to electrostatic repulsion with DNA, which allows recruitment of chromatin remodeling factors and increases gene transcription [9,10]. DNA-dependent PARPs act as DNA damage sensors involved in DNA break repair, with ADP-ribosylation at the double-stranded breaks acting as a signal, which allows the recruitment of DNA repair enzymes to the lesion site [11]. In cases of major DNA damage, overactivation of PARP1 can induce a depletion of the NAD^+^ pool in the cell, inhibiting ATP production and cellular metabolism, ultimately leading to cell death by necrosis. In a final example, PARP5a and -5b bind and ADP-ribosylate the telomeric repeat factor 1 (TRF1), reducing its binding ability to DNA and upregulating telomere maintenance [12].

In addition to these physiological functions, recent studies have highlighted the role of PARPs as actors in host antiviral response. In the context of viral infections, PARP expression can be induced, as reported for PARP3, -4, -5a, -5b, -7, -8, -9, -10, -11, -12, -13, and -14, in cells infected with coronaviruses [13,14]. Some PARPs are considered as interferon-stimulated genes (ISGs) and can consequently play a key role in the regulation of the innate immune response. Their antiviral activity can either be direct, by interfering with the critical steps of the viral replication cycle, or indirect, through immunomodulatory mechanisms. This review will summarize and discuss the direct and indirect antiviral properties of PARPs as well as the mechanisms brought into play by viruses to escape them.

## 2. Inhibition of Viral Cycle Steps by the PARPs 

Every step of the viral cycle, from entry into to exit from the infected cell, represents a potential target for antiviral proteins. PARPs have been shown to target several steps of the virus replication cycle, mostly by inhibiting viral genome transcription and translation (Figure 1).

### 2.1. Degradation of Viral Nucleic Acids

In 2002, a screening of antiviral molecules in mammal cell showed the first evidence of the antiviral activities of PARPs. PARP13, initially called zinc-finger antiviral protein (ZAP), was identified as an inhibitor of retrovirus replication in rat cells [15]. This antiviral activity of rat ZAP was first discovered using a truncated PARP13 protein, which consists of only one of the four CCCH-type zinc-finger domains that mediate RNA binding and did not include the C-terminal PARP-like domain. Several isoforms of PARP13 have since been described in rats and humans [16,17]. Northern blots of the cytoplasmic fractions from infected cells expressing rat PARP13 showed a specific degradation pattern of the cytoplasmic viral RNA, while viral RNA within the nucleus remained intact [15]. Then, experiments of overexpression and downregulation in both rat and human cells showed that PARP13 possessed broad antiviral activity against a wide range of viral species, including alphaviruses, filoviruses (Ebola virus and Marburg virus), xenotropic murine leukemia virus-related virus (XMRV), coxsackievirus B3, Japanese encephalitis virus, human immunodeficiency virus 1 (HIV-1), Sindbis virus (SINV), hepatitis B virus (HBV) and influenza A virus (IAV) [18,19,20,21,22,23,24,25,26]. PARP13 targets cytoplasmic viral RNA, preventing transcription and translation of the viral genome. Interestingly, no impact of PARP13 overexpression on herpes simplex virus 1, vesicular stomatitis virus (VSV), yellow fever virus, poliovirus, dengue virus or Zika virus (ZIKV) replication has been found, proving that PARP13 antiviral properties are specific to certain viral species [18,24].

The mechanism of PARP13 antiviral activity was then unraveled. PARP13 forms a homodimer, which can bind viral RNA through its four zinc-finger motifs [27,28]. At the molecular level, it was recently shown that PARP13 N-terminal domain specifically binds to CpG dinucleotides in single-stranded RNAs, the interaction being further strengthened by additional guanine and cytosine [29]. Following the binding of PARP13 to viral RNA, recruitment of the exosome to the target RNA is induced leading to its degradation. Coimmunoprecipitation assays showed a direct interaction between rat PARP13 and the exosome component hRrp46p, while human PARP13 interacted with hRrp42, allowing exosome complex formation [21,30]. Recent studies have suggested that, even though some viruses have strongly CpG suppressed genomes, they can be restricted by PARP13 as it seems that, more so than their number, the localization of the CpG motifs is crucial for this restriction [31,32]. 

Furthermore, RNA helicases have been shown to be involved in the antiviral activity of PARP13. In 2007, Margaret and colleagues showed a synergistic inhibitory effect against SINV of PARP13 and an unknown ISG [33]. This PARP13 cofactor was then identified as p72 DEAD box RNA helicase. Since an inhibition of its helicase activity led to a diminution of RNA degradation, it was suggested that p72 facilitates PARP13-mediated exosome degradation by unwinding the viral complex RNA tertiary structure [34]. In a similar way, the DEXH box RNA helicase, DHX30, was shown to be a PARP13-interacting protein required for optimal antiviral activity [35]. During the HIV-1 replication cycle, PARP13 allows recruitment of the decapping complex to the 5′-end of viral RNAs, preventing their classical cap-dependent translation initiation and leading to their degradation [21]. PARP13 can also recognize CpG motifs in retroviral RNA and recruits host factors, including the endonuclease KHNYN, which then degrade viral RNA [36]. Finally, PARP13 has been shown to interact with poly(A)-specific ribonuclease (PARN) to remove the poly-(A) tail of HIV-1 RNA, thereby exposing the 3′-end to the exosome and leading to RNA degradation.

Even if the antiviral properties of PARP13 are now well-known, there is still a discussion about which of its two major isoforms displays the most efficient antiviral activity. PARP13 exists in two major isoforms: a long, constitutive full-length isoform (named ZAP-L or PARP13L) and an interferon-inducible short isoform (ZAP-S, PARP13S) lacking the C-terminal PARP domain [18]. Evolutionary analysis showed a positive selection confined to the PARP C-terminal domain, indicating that it would be an important interface for PARP13 interaction with the genome of constantly mutating viruses [16]. In this study, it was found that ZAP-L was a more potent inhibitor than ZAP-S of murine leukemia virus (MLV) and Semliki forest virus replication by 2 to 3-fold. Another study also showed that the PARP-like domain plays a crucial role in ZAP-L’s antiviral activity against IAV [37]. By comparing both PARP13 isoforms, ZAP-L was recently shown to be a primary antiviral effector against the alphavirus SINV [38]. The isoform-specific targeting of viral RNA was due to differences in the subcellular localization of the two isoforms, which were mediated by the presence or absence of a C-terminal prenylation motif and allowed the recruitment of ZAP-L to sites of SINV RNA replication at the plasma membrane and in endolysosomes. Even if the PARP-like domain of ZAP-L lacks the catalytic activity of functional PARPs, mutations in this domain leads to the loss of its antiviral activity, indicating an essential function of the PARP-like domain in restricting alphaviruses in humans [39].

PARP7 might use a similar mechanism to degrade genomic RNA of viruses belonging to *Togaviridae* family (SINV and rubella virus). Immunoprecipitation assays have shown that PARP7 binds SINV RNA as well as the exosome complex component 5 (EXOSC5) via its N-ter CCCH-type zinc-finger domain [40]. During SINV infection, reactive oxygen species are produced by damaged mitochondria and induce oxidation of the nucleoporin protein Nup62, leading to a cytoplasmic accumulation of PARP7 where it binds viral RNA, thereby promoting its degradation.

### 2.2. Inhibition of Viral Replication

PARP1, -2 and -9 are known to be involved in transcription regulation in the cell. They can act as scaffold proteins, modifying the chromatin structure, and then facilitating or preventing the binding of transcription factors to DNA [41]. Their ability to modulate cell mRNA transcription can also affect viral RNA transcription. Since viruses rely on cellular machinery for their replication, cells have developed many defense mechanisms to prevent this, some of them involving nuclear PARPs.

In infected cells, the Epstein-Barr virus (EBV) maintains latency in the form of circular double-stranded (ds) DNA, replicating its genome once per cell cycle. This replication is dependent on the viral protein EBNA1 binding to the dsDNA at the origin of plasmid replication (OriP) site. Several studies have shown that PARP1 or PARP5 were also binding partners of OriP, competing with EBNA1, and causing a decrease in EBV DNA replication and maintenance in latently infected cells [42]. Conversely, inhibition of PARP activity increased maintenance of OriP plasmids, while inhibitors of OriP replication were also stimulators of PARP activity [42]. Both PARP1 and PARP5 were shown to interact with and modify EBNA1, remodeling protein–DNA structure and leading to negative regulation of OriP replication [43,44]. 

In the same way, and in a sequence-specific manner, PARP1 binds the replication origin TR of the Kaposi sarcoma-associated herpesvirus (KSHV). Moreover, PARP1 catalyzes poly-ADP-ribosylation of the latency-associated nuclear antigen (LANA), affecting maintenance of the viral genome in the latently infected cells [45]. Taken together these results showed a similar inhibition mechanism of latency maintenance of two γ-*herpesvirinae* viruses, KSHV and EBV, by PARP1.

In addition to disrupting herpesvirus maintenance in the cell, PARP1 interferes with their reactivation. Lupey-Green and colleagues showed that PARP1 specifically bound the EBV lytic promoter BZLF1, inhibiting viral reactivation. The viral protein Zta, which also binds BZLF1, is able to antagonize the PARP1-mediated inhibition of EBV lytic reactivation [46]. Moreover, PARP1, synergistically with the Ste20-like kinase hKFC, PARylates the KSHV protein replication and transcription activator (RTA), which has a central role in the switch between latency and lytic cycle. Interaction of the PARP1/hKFC complex with RTA decreases its recruitment to promoter regions and disrupts RTA-mediated KSHV reactivation [47].

Viral transcription inhibition by PARPs also occurs through epigenetic modifications. This mechanism was reported as mediating the silencing of retrotransposable elements and inhibiting transcription of integrated retroviruses. An initial study conducted in 2005 showed that PARP1 bound HIV-1 TAR RNA. TAR is the binding site of the trans-activator of HIV-1 LTR Tat-containing complex p-TEFb, which elongates the HIV-1 RNA. PARP1 binds TAR with a higher affinity than the p-TEFb complex, leading to p-TEFb displacement from HIV-1 RNA, suppressing Tat-mediated transcriptional elongation [48]. This PARP1-mediated retrovirus transcription inhibition was also shown to be efficient against MLV. It is mediated by epigenetic mechanisms that involve DNA methylation and histone deacetylation, and does not seem to require the catalytic activity of PARP1 [49]. PARP1-mediated retroviral silencing also occurs before integration in a catalytic-independent manner. It has been shown that PARP1 can repress retroviruses prior to viral DNA integration by mechanisms involving histone deacetylases but not heterochromatin formation [50]. Finally, in addition to its role in the inhibition of transcription of integrated proviral DNA, PARP1 can interfere with HIV-1 integration into cellular DNA, but this point remains controversial [51,52,53,54]. 

Otherwise, it has been suggested that PARP12 inhibits steps before VSV replication and secondary transcription [55].

### 2.3. Translation Inhibition

Translation of the viral genome is a usual target of ISGs in order to block viral replication. Gao and colleagues showed that PARP13 could also inhibit HIV-1 translation. Overexpression of human PARP13 in 293TRex cells reduced production of the HIV-1 Nef protein that plays an important role in virus replication *in vivo* by nearly 13-fold, whereas, in the same condition, nef mRNA expression decreased only by 5-fold, suggesting an inhibition of mRNA translation. Moreover, translation inhibition by PARP13 was not related to mRNA degradation. Polysome profiling, which analyzes association of mRNA with ribosomes, has confirmed that PARP13 excludes the target mRNAs from polysome fractions without affecting global cell translation [56]. The same study found that PARP13 interacted with eukaryotic initiation factor 4-A (eIF4)-A, thereby competing with eIF4-G for binding and stopping the formation of the canonical translation initiation complex on viral mRNA. Otherwise, the E3 ligase tripartite motif-containing protein 25 (TRIM25) was recently found to be a novel PARP13 cofactor involved in viral translation inhibition through an ubiquitination-dependent mechanism targeting unidentified host proteins [57,58]. PARP13 has also been shown to inhibit the formation of the translation initiation complex on IAV mRNA in a PARP-domain independent manner [23]. Finally, although this question has not yet been elucidated, it seems logical that the decapping of HIV-1 mRNA mediated by PARP13 would impair the translation initiation process since most of HIV-1 translation occurs in a cap-dependent manner [21,59].

Though a majority of studies on the translation inhibition mediated by the PARPs focus on PARP13, recent discoveries have shown that PARP7, -10 and -12 are likewise inhibitors of viral translation [60]. PARP12 exhibits antiviral activity against many viruses including Venezuelan equine encephalitis virus (VEEV), VSV, SINV, encephalomyocarditis virus (EMCV), rift valley fever virus (RVFV) and Chikungunya virus (CHIKV) [61]. It is interesting to note that PARP10 and PARP7 overexpression also inhibit VEEV replication, which may explain why a downregulation of PARP12 does not suppress the antiviral state of the cell due to a redundancy of the roles between PARPs [60]. PARP12 shares a very similar structure with PARP13 consisting of zinc-finger motifs as well as the existence of two isoforms, with the short-one lacking the PARP domain. However, contrary to PARP13, only the long isoform displays antiviral activity, and while PARP13 is catalytically inactive, PARP12 is capable of MARylating proteins [62]. PARP12′s catalytic activity has been shown to be necessary to downregulate mRNA translation [62]. 

### 2.4. Targeting Viral Proteins

Despite its PARP signature, PARP9 was initially thought to be devoid of catalytic activity, which allows ADP-ribosylation [63]. In 2003, the protein Deltex E3 ubiquitin ligase 3L (DTX3L) was identified as a PARP9 binding partner [64]. Unlike PARP9 alone, the heterodimer PARP9/DTX3L displays mono-ADP-ribosylating activity. MARylation of a protein by PARP9 leads to its ubiquitination by DTX3L causing proteosomal degradation of the target protein [65]. The PARP9–DTX3L complex is thereby responsible for the degradation by ubiquitination of many viral proteins such as the 3C proteases of EMCV and human rhinovirus, both of which are viruses belonging to the *Picornaviridae* family [66]. Interestingly, this does not occur with the respiratory syncytial virus NS1 protein, suggesting that ubiquitination and degradation mediated by DTX3L could be specific to *Picornaviridae* 3C proteases [66]. The long isoform of PARP13, ZAP-L, which does not harbor an ADP-ribosylation activity, has been shown to bind the PARylated IAV polymerase proteins PA and PB2, leading to their ubiquitination and proteasomal degradation [37]. 

An avian influenza virus (AIV) protein has been shown to be targeted by PARP10; this interaction plays a regulatory role in virus replication. Expression of AIV NS1 protein in infected cells causes relocalization of PARP10 from the cytoplasm to the nuclei and reduces endogenous PARP10 expression [67]. In addition, decrease in PARP10 expression leads to cell proliferation and promotes viral replication [67]. 

Finally, in addition to its ability to suppress viral translation, PARP12 restricts ZIKV infection through degradation of viral proteins. PARP12L has been shown to be able to MARylate the nonstructural ZIKV proteins NS1 and NS3, both of which are involved in viral replication and host immune response modulation. NS1 and NS3 MARylation leads to their PARylation, presumably by another member of the PARP family, which increases their Lys 48 ubiquitination and subsequent proteasomal degradation [68]. 

## 3. Immunomodulation

The role of PARPs in immune response has been investigated in several studies, and reviewed in [69,70,71]. Given that the proinflammatory roles of PARP1, through triggering of NF-κB signaling pathway, induction of chemokine expression and activation of immune cells such as neutrophils, macrophages and dendritic cells, has been extensively described, the immunomodulation properties of PARP1 have been discussed in several other reviews and will not be discussed here [72,73,74]. However, several other members of the PARP family, including PARP7, -9, -10, -11, -12, -13 and -14, have important roles in innate immunity, and the main ones are summarized here (Figure 2).

PARP11-induced MARylation of the β-transducin repeat-containing protein (β-TrCP), an E3 ubiquitin ligase, has been shown to promote ubiquitination and subsequent degradation of interferon alpha/beta receptor 1 (IFNAR1) [75]. ZAP-S, the short isoform of PARP13, also has immunomodulatory properties [38,76]. Its overexpression in human HEK293Y cells enhances IFN-β mRNA expression and oligomerization of the viral RNA sensor retinoic acid-inducible gene I (RIG-I), which leads to robust activation of downstream antiviral signaling through the interferon regulatory factor 3 (IRF3) pathway [76]. On another track, a few studies have suggested that PARP7, -10 and -11 may downregulate the inflammatory response. PARP7 has been shown to mono-ADP-ribosylate TANK-binding kinase 1 (TBK1) downstream of the RIG-I signaling pathway [77]. Consequently, the TBK1-induced dimerization of IRF3 decreases, leading to impaired IFN production. PARP10 exercises negative feedback on the NF-κB signaling pathway through MARylation of NF-κB essential modulator (NEMO), thereby preventing its K63 polyubiquitination [78]. PARP12 could also enhance the signaling cascade leading to NF-κB activation [62]. Moreover, it was recently shown that ZAP-S binds to and mediates the degradation of several host IFN mRNAs, thereby acting as a negative feedback regulator of the interferon response [38]. 

PARP9, -12, -13 and -14 have been reported to enhance the cell innate immune response. Increased concentrations of PARP9 and DTX3L, or of the E3-ubiquitin ligase complex formed by these two proteins, have been observed in cells stimulated with IFN-γ. PARP9 and DTX3L expression depends on the same promoter, which is bidirectional and contains binding sites for STAT-1 and IRF1, both of which are transcriptional factors involved in the antiviral response [79]. IFN-induced overexpression of the IFN-signaling molecule STAT-1 leads to upregulation of the translation of PARP9 and DTX3L. In turn, the PARP9–DTX3L complex is a direct binding partner of STAT-1, enhancing STAT-1 phosphorylation and, consequently, its activation. This binding also promotes STAT-1 nuclear relocalization, increasing the transcription efficiency of ISGs, thereby leading to amplification of the innate immune response [66]. In addition, the PARP9–DTX3L complex catalyzes ubiquitination of a subset of histones from the host, notably H2BJ, increasing histone methylation, which leads to chromatin remodeling and, once again, to increased transcription of ISGs. 

Similarly, a study conducted in human primary macrophages showed PARP14 to be specifically bound to STAT-6 responsive promoters, preventing STAT-6-mediated transcription. However, in the presence of the anti-inflammatory cytokine IL-4, the MARylation activity of PARP14 is triggered, leading to its own MARylation by an autocatalytic process that allows binding of STAT-6 and transcription activation [80]. PARP14 thereby mediates a switch between repression and activation of ISG transcription depending on the presence or absence of IL-4. 

Interestingly, functional redundancy between PARP9 and PARP14 has been hypothesized due to structural similarities. Following macrophage stimulation with IFN-γ, it was shown that PARP9 and PARP14 are both upregulated but then assume opposite roles. PARP9 potentiates the response to IFN-γ by enhancing phosphorylation of STAT-1 whereas PARP14, by ADP-ribosylating STAT-1, decreases the response while increasing phosphorylation of STAT-6, thereby promoting the anti-inflammatory response mediated by IL-4 [81]. It is interesting to note that PARP9 could inhibit MARylation of STAT-1 by PARP14, restoring the proinflammatory response of the cell. In addition, PARP9 binds the STAT-6 responsive promoter in replacement of PARP14 but without triggering the same switch between activation and repression of transcription, since STAT-6-dependent transcription is still impaired in PARP14 -/- primary macrophages despite IL-4 stimulation. 

Finally, in addition to ISG transcription regulation, PARP14 directs T cell towards Th2 differentiation by promoting binding of STAT-6 to the *Gata3* promoter [82,83]. 

All in all, PARPs are ISGs acting as positive or negative regulators of the inflammatory process induced during viral infection. Even without direct antiviral effects on the virus replication cycle, they can modulate host defenses, inducing either upregulation of the cell antiviral state to fight viral infection or downregulation of the immune response to prevent inflammatory damages.

## 4. Strategies to Escape the Antiviral Activities of PARPs

Through numerous studies, it is now obvious that PARP-induced PARylation and MARylation are important post-translational modifications leading to activation of cell innate immune factors mobilized against viral infection. Viruses have developed many ways to counteract PARP antiviral activities. The genome of several RNA viruses such as coronaviruses (severe acute respiratory syndrome-related coronavirus (SARS-CoV)-1 and -2, Middle East respiratory syndrome coronavirus (MERS-CoV)), alphaviruses (CHIKV and SINV), hepatitis E virus (HEV) and the murine hepatitis virus (MHV) encodes macrodomain-containing proteins. A macrodomain is a highly conserved sequence of 170 to 180 amino-acids, originally found in human core histones macroH2A, which is a component of chromatin [84]. Macrodomains are capable of recognizing and binding mono- and poly-ADP-ribosylated proteins, subsequently modifying the ADP-ribose residues and downstream signaling or antiviral roles [85,86,87]. Alphavirus and coronavirus macrodomains were thus shown to be essential to efficient virus replication and virulence in cells [88,89,90]. Interestingly, PARP14 exhibits macrodomains, allowing them to remove ADP-ribose from substrates and exercise regulation or feedback on their PARylation and MARylation activities [80]. 

Moreover, some viral proteins can directly interact with PARPs, thereby neutralizing their antiviral properties. A study conducted on IAV showed that the viral protein NS1 antagonizes PARP13-induced mRNA decay by reducing its RNA-binding capacity. In addition to this direct interaction, IAV NS1 protein can alter PARP transcription. Taken together, these results explain why only prior to viral protein expression can PARP13 alter IAV replication [23]. PARP13 is also responsible for inhibition of enterovirus (EV) A71replication [91,92]. A recent study suggests that 3C protease cleaves PARP13 at the Gln-369 residue, leading to the loss of this inhibitory activity on the viral replication [93]. Regarding DNA viruses, the RTA protein of the murine γ-herpesvirus 68, which is a reactivator of the lytic viral cycle, prevents the formation of the PARP13 homodimer required for viral RNA recognition and degradation [94]. The KSHV processivity factor (PF)-8 protein, through direct interaction, leads to PARP1 proteasomal degradation, disrupting its inhibition of lytic cycle reactivation [95]. Finally, HSV has been shown to lead to hyperphoshorylation, nuclear transportation and retention by infected cell protein 0 (ICP0), an immediate early viral protein, and to proteosomal degradation of PARP5a, thereby enhancing HSV replication [96]. Another HSV protein, UL41, which is a tegument protein involved in host mRNA degradation, has been shown to be involved in PARP13L mRNA degradation leading to increased viral replication [97]. 

Recent studies have reported that an excessive activation of PARP1 can induce cell death, defined as parthanatos, related to DNA damage signaling. This phenomenon is also observed during viral infections where PARP1 is cleaved and therefore activated [98]. It also occurs in ZIKV infected cells through direct interaction of the helicase NS3 with PARP1, resulting in the death of neurons from infected brain [99]. However, if PARP1-mediated cell death is involved in the pathogenicity and the clinical symptoms caused by the viral infection, it could also play an important role in eliminating intracellular pathogens. In order to preserve virus replication, the adenovirus E4orf4 protein associates with PARP1, reducing its phosphorylation on serine residues, reported to enhance its activity, and therefore preventing its excessive activation [100].

## 5. Conclusions

Current knowledge about PARPs, at the interface between host and viruses, seems to point to a globally underestimated and important role of this family of enzymes in host defense. Due to their cytoplasmic and nuclear localization, PARPs are capable of inhibiting virus life cycle at several stages, from transcription to translation. In addition, PARPs can act indirectly by stimulating host innate immunity through activation of intracytoplasmic sensors of pathogen-associated molecular patterns and ISG transcription. To date, ten of the seventeen human PARPs have been shown to display antiviral properties. The other seven remain to be studied and could possibly also exhibit antiviral activities. 

PARPs represent potentially interesting targets for new antiviral therapeutics since PARP agonists could help to restore the antiviral state of an infected cell. Although upregulation of one PARP may not be sufficient for control of the infection, the targeting of multiple PARPs, some for their direct antiviral activity and others for their immunostimulating properties, might be an interesting strategy that remains to be explored. 

## Figures and Tables

**Figure 1 viruses-13-00582-f001:**
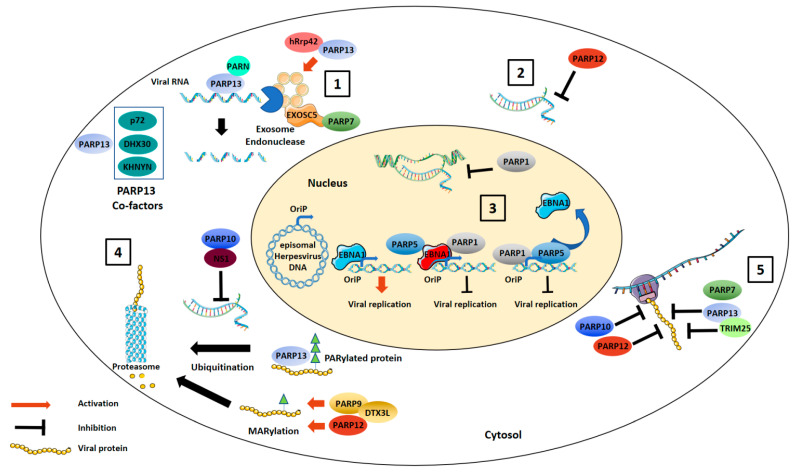
Direct antiviral activities of PARPs. PARP13 and PARP7 can induce exosome-mediated degradation of target viral RNAs (**1**). PARPs can also inhibit viral genome replication. PARP12 inhibits viral RNA transcription within the cell cytoplasm (**2**). PARP1 or PARP5 PARylate or directly interact with EBNA1, preventing EBNA1 binding to the OriP promoter and inhibiting Epstein–Barr virus (EBV) replication (**3**). PARPs directly interact with viral proteins. PARP9/DTX3L complex and PARP12 catalyze mono-ADP-ribosylation (MARylation) of viral proteins leading to their proteosomal degradation while PARP10, through binding to avian influenza virus NS1, prevents viral RNA replication. PARP13 binds already PARylated influenza A virus proteins leading to their proteosomal degradation (**4**). Finally, PARP7, -10, -12 and -13 are inhibitors of viral translation stopping the formation of the translation initiation complex on viral mRNA (**5**).

**Figure 2 viruses-13-00582-f002:**
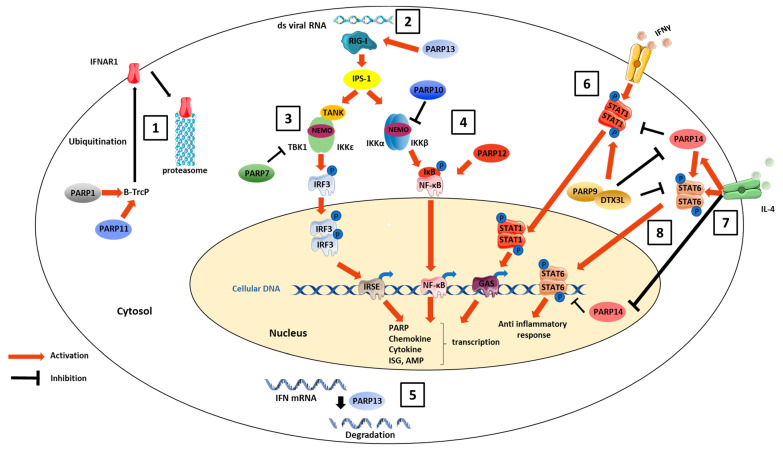
Indirect antiviral activities of PARPs through immunomodulation. PARP1 and PARP11-induced MARylation of the β-transducin repeat-containing protein (β-TrCP), an E3 ubiquitin ligase, promotes interferon alpha/beta receptor 1 (IFNAR1) ubiquitination and degradation (**1**). Concerning RNA virus sensing pathways, PARP13 overexpression enhances oligomerization of retinoic acid-inducible gene I (RIG-I) (**2**), which leads to robust activation of downstream antiviral signaling through the interferon regulatory factor 3 (IRF3) pathway. On the other hand, PARPs can also downregulate the antiviral defenses. PARP7 MARylates TANK-binding kinase 1 (TBK1), decreasing IRF3 activation and leading to impaired IFN production (**3**). Otherwise, PARP12 enhances the signaling cascade, leading to NF-κB activation, whereas PARP10 exerts negative feedback on this pathway through MARylation of NF-κB essential modulator (NEMO) (**4**). PARP13 mediates the degradation of several host IFN mRNAs (**5**), thereby acting as a negative feedback regulator of the interferon response. The PARP9–DTX3L complex is a direct binding partner of STAT-1, promoting STAT-1 phosphorylation and nuclear relocalization, thereby increasing ISG transcription and leading to amplification of the innate immune response (**6**). PARP14 specifically binds to STAT-6 responsive promoters (**7**), preventing STAT-6-mediated transcription. The inflammatory environment influences PARP9–DTX3L and PARP14 immunomodulatory properties. In the presence of the anti-inflammatory cytokine IL-4 (**8**), PARP14 carries out its own MARylation by an autocatalytic process that allows binding of STAT-6 and transcription activation. PARP9–DTX3L potentiates the response to IFN-γ by enhancing phosphorylation of STAT-1, whereas PARP14, by ADP-ribosylating STAT1, decreases the response while increasing phosphorylation of STAT-6, thereby promoting the anti-inflammatory response mediated by IL-4. PARP9–DTX3L can in turn inhibit MARylation of STAT-1 by PARP14.

**Table 1 viruses-13-00582-t001:** Overview of the 17 human Poly-adenosine diphosphate (ADP)-ribose polymerases (PARPs) including alternative names, structural characteristics, antiviral activity, if any, and targeted viruses.

Name	Other Names	PARP Activity	PARP Subfamilies	Characteristic Domains	Antiviral Activity	Viruses Targeted
PARP1	ARTD1	PARylation	DNA-dependent PARPS	BRCT, WGR	Transcription and replication inhibition	EBV, HIV, KSHV, MLV
PARP2	ARTD2	PARylation	DNA-dependent PARPS	WGR	ND	ND
PARP3	ARTD3	PARylation	DNA-dependent PARPS	WGR	ND	ND
PARP4	ARTD4 KIAA0177	MARylation	Unclassified	BRCT	ND	ND
PARP5a	ARTD5 TANK1 TIN1	PARylation	Tankyrases	ANK	Replication inhibition	EBV
PARP5b	ARTD6 TANK2 TNKL	PARylation	Tankyrases	ANK	Replication inhibition	EBV
PARP6	ARTD17	MARylation	Unclassified	HPS	ND	ND
PARP7	ARTD14 TIPARP	MARylation	CCCH PARPs	Zinc-fingers, WWE	Replication and translation inhibition	SINV, Rubella virus, VEEV
PARP8	ARTD16	MARylation	Unclassified	HPS	ND	ND
PARP9 +DTX3L	ARTD9 BAL1	MARylation	MacroPARPs	Macrodomains	Viral protein degradation	EMCV
PARP10	ARTD10	MARylation	Unclassified	UIM	Transcription and replication inhibition Viral protein degradation	AIV, VEEV
PARP11	ARTD11	MARylation	Unclassified	WWE	ND	ND
PARP12	ARTD12 ZC3HDC1	MARylation	CCCH PARPs	Zinc-fingers, WWE	Transcription and replication inhibition Viral protein degradation	CHIKV, EMCV, RFVF, SINV, VEEV, VSV
PARP13	ZAP ARTD13 ZC3HDC2	Inactive	CCCH PARPs	Zinc-fingers, WWE	Replication and translation inhibition Viral RNA and protein degradation	HIV, IAV, HBV, SINV, XMRV, Ebola virus, Marburg virus, MHV68
PARP14	ARTD8 BAL2	MARylation	MacroPARPs	Macrodomains, WWE	ND	ND
PARP15	ARTD7 BAL3	MARylation	MacroPARPs	Macrodomains	ND	ND
PARP16	ARTD15	MARylation	Unclassified	TMD	ND	ND

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
