# Peer review of "The Antiviral Activities of Poly-ADP-Ribose Polymerases"

_viruses, 2021, doi:10.3390/v13040582_

Round 1

Reviewer 1 Report

Malgras and colleagues provide an up to date and fairly detailed review of the published roles of various PARPs in certain viral replication cycles.  This remains an important topic and the authors contribution will assist researchers in differing viral fields to readily obtain these data.  The authors should consider the following to strengthen the contribution.

  1. The authors tend to organize the content by PARPs, often chronologically from PARP1 to PARP16 (starting in Table 1).  Also, the authors switch between virus groups, often in the same paragraph which many readers may find confusing. It might be beneficial if the review was organized around the general biological properties of the virus groups instead.  For example, a discussion could occur around those viruses whose replication includes a DNA phase and occurs in the nucleus, as these all seem to involve similar PARPs. Conversely, RNA viruses that do not have a DNA phase and solely replicate in the cytoplasm could be compared.  Do the subcelluar localizations of the relevant PARPs make sense in these contexts?  The goal should be to correlate the PARP actions with the viral biology. Otherwise, readers are left wondering why PARPs do not degrade DNA virus RNAs, etc.
  2. Additional description of the history of ZAP being PARP13 would be helpful as much of the virus literature prefers ZAP.
  3. The authors should try to extend the review published by Fehr et al. in 2020 (ref.99).  For example, the previous authors note that the HSV ICP0 and UL41 proteins regulate PARylation.  These could be included in the holistic approach suggested above.
  4. The authors seems to believe that PARPs are only involved in inhibiting virus replication.  When the data might not fit this scenario, they state things like "PARP13 antiviral properties are specific to certain viruses" (line 102).  They should consider the possibility that viruses might utilize PARPs to benefit their own replication.  It might be helpful for them to review the older literature where viral regulatory proteins from SV40, AdV, and HSV were associated with PARPs.
  5. Another area that the authors could consider discussing is why PARP1 is cleaved during virus induced programmed cell death.  At least they might acknowledge that that occurs.

Author Response

Reviewer 1 :

Malgras and colleagues provide an up to date and fairly detailed review of the published roles of various PARPs in certain viral replication cycles.  This remains an important topic and the authors contribution will assist researchers in differing viral fields to readily obtain these data.  The authors should consider the following to strengthen the contribution.

  1. The authors tend to organize the content by PARPs, often chronologically from PARP1 to PARP16 (starting in Table 1).  Also, the authors switch between virus groups, often in the same paragraph which many readers may find confusing. It might be beneficial if the review was organized around the general biological properties of the virus groups instead.  For example, a discussion could occur around those viruses whose replication includes a DNA phase and occurs in the nucleus, as these all seem to involve similar PARPs. Conversely, RNA viruses that do not have a DNA phase and solely replicate in the cytoplasm could be compared.  Do the subcelluar localizations of the relevant PARPs make sense in these contexts?  The goal should be to correlate the PARP actions with the viral biology. Otherwise, readers are left wondering why PARPs do not degrade DNA virus RNAs, etc.

The description of the antiviral activities of PARPs is already organized according to the major stages of the virus replication cycle (inhibition of transcription, translation, etc.) in which DNA and RNA viruses are differentiated. However, the separation between RNA and DNA viruses was introduced in the chapter “Strategies to escape the antiviral activities of PARPs”  in order to make it easier to read.

  1. Additional description of the history of ZAP being PARP13 would be helpful as much of the virus literature prefers ZAP.

This has been completed:” This antiviral activity of rat ZAP was first discovered using a truncated PARP13 protein, which consisted in only one of the four CCCH-type zinc-finger domains that mediate RNA binding and did not include the C-terminal PARP-like domain. Several isoforms of PARP13 have since been described in rats and humans”

  1. The authors should try to extend the review published by Fehr et al. in 2020 (ref.99).  For example, the previous authors note that the HSV ICP0 and UL41 proteins regulate PARylation.  These could be included in the holistic approach suggested above.

This has been added in the “strategies to escape the antiviral activities of PARPs” paragraph:” HSV has been shown to lead to hyperphoshorylation, nuclear transportation and retention by infected cell protein 0 (ICP0), an immediate early viral protein, and to proteosomal degradation of PARP5a, thereby enhancing  HSV replication [95]. Another HSV protein, UL41, which  is a tegument protein involved in host mRNA degradation, has been shown to be involved in PARP13L mRNA degradation, leading to increased viral replication [96]. »

  1. The authors seems to believe that PARPs are only involved in inhibiting virus replication.  When the data might not fit this scenario, they state things like "PARP13 antiviral properties are specific to certain viruses" (line 102).  They should consider the possibility that viruses might utilize PARPs to benefit their own replication.  It might be helpful for them to review the older literature where viral regulatory proteins from SV40, AdV, and HSV were associated with PARPs.

The review focuses on the antiviral activities of the PARPS, nonetheless, this is discussed in the “strategies to escape the antiviral activities of PARPs” paragraph.

  1. Another area that the authors could consider discussing is why PARP1 is cleaved during virus induced programmed cell death.  At least they might acknowledge that that occurs.

PARP1 cleavage during viral infections has been added to the review in the “strategies to escape the antiviral activities of PARPs” paragraph as well as the mechanism used by adenoviruses to prevent it.

Reviewer 2 Report

Viruses – Review – by Mathilde Malgras et al

“The antiviral activities of poly-ADP-ribose polymerases”

General overview:

The authors in this manuscript described a detailed information about functions and different protein interactions of various poly-ADP-ribose polymerases PARPs with cellular and viral proteins. The title of the manuscript clearly describes the manuscript and fits the purpose. The content was also fulfilled with many reference studies.

Review:

The manuscript is well written and showed a comprehensive understanding of the roles of PARPs that either counteract viral infections or enhance innate immunity protein expression to fight the viral infection. A summary list of all known PARPs and their characteristics was helpful as a reference which was then followed by detailed description of the involvement of PARPs in targeting proteins in different viral life cycle. This detail give more insight into how these PARPs target viral nucleic acid, prevention of viral transcription and translation and how these PARPs can lead to degradation of viral proteins. Another aspect of the function of these PARPs are their roles in enhancing innate immunity by interacting with various proteins that are responsible for different immunomodulatory pathways. Finally, a short summary of how some viruses possess macrodomain that could reverse critical PARPs functions. However, below are some comments to this work

Comments:

  1. The symbol NAD +; should be superscripted (NAD+) to be consistent with the literature.
  2. Table 1: AIV acronym refer to avian Influenza virus! Should it be in general Influenza A virus (IAV).
  3. Table 1: PARP10 viruses targeted; there are other viruses should be included as well.
  4. Line 210: the sentence needs to be rephrased to clarify what was depleted from these cells.
  5. Line 237: the significant of the last sentence is not clear and why the this binding leads to translation inhibition.
  6. Line 250: a spelling mistake “ arbor ” to “harbor”.
  7. Line 255: the word “There” should be “Therefore”.
  8. Line 256: the word “bounds” should be “bound” and I suggest to rewrite the sentence.
  9. Line 265: the acronym “K-48” should be defined “Lys 48” or anything else.
  10. Line 270: the sentence “…., the immunomodulation properties of PARP1 will not be discussed here [71-73]” should be modified to indicate that these review references discussed the properties of PARP1 and will not discussed in our review.
  11. Figure 2:
    1. Define the arrows and stop lines and what do they indicate to.
    2. Step number 3 is missing in the figure.
  12. Line 295: this long paragraph should be broken into shorter paragraphs and order the paragraphs to follow figure 2 numbering or re-order the numbering in figure 2 to that in the text.
  13. Line 327-329: remove this sentence as it is not related to viruses.
  14. Line 334: the sentence started with ( In contrast, ….. ) should be re-written using present tense.
  15. Line 343 and 344: would be better to keep the term MARylation consistent, choose either mono-ADP-riboylation or MARylation after explaining the term.
  16. Line 357: the genome of SARS-CoV-2 also has this domain and should be included in to the list of viruses.
  17. Line 364: “ removing ADP-ribose residues “ not very accurate, macrodomain removes the modification from certain residues but not the residue itself.
  18. Line 369: this study referred to as reference [75] discussed PARP14 only and speculate the same activity to some other PARPs. Therefore, it would be better to be specific in this.
  19. Line 373: the acronym “AIV” should be “IAV” for Influenza A virus.
  20. Line 374: space between “NS1” and “antagonizes”.
  21. Line 377: extra word “can”.
  22. Line 378: this sentence “ This viral protein …. “ should be re-written because of some grammatical errors.

Author Response

Reviewer 2 :

Viruses – Review – by Mathilde Malgras et al

“The antiviral activities of poly-ADP-ribose polymerases”

General overview:

The authors in this manuscript described a detailed information about functions and different protein interactions of various poly-ADP-ribose polymerases PARPs with cellular and viral proteins. The title of the manuscript clearly describes the manuscript and fits the purpose. The content was also fulfilled with many reference studies.

Review:

The manuscript is well written and showed a comprehensive understanding of the roles of PARPs that either counteract viral infections or enhance innate immunity protein expression to fight the viral infection. A summary list of all known PARPs and their characteristics was helpful as a reference which was then followed by detailed description of the involvement of PARPs in targeting proteins in different viral life cycle. This detail give more insight into how these PARPs target viral nucleic acid, prevention of viral transcription and translation and how these PARPs can lead to degradation of viral proteins.

Another aspect of the function of these PARPs are their roles in enhancing innate immunity by

interacting with various proteins that are responsible for different immunomodulatory pathways.

Finally, a short summary of how some viruses possess macrodomain that could reverse critical PARPs

functions. However, below are some comments to this work

Comments:

  1. The symbol NAD +; should be superscripted (NAD+) to be consistent with the literature.

This has been corrected.

  1. Table 1: AIV acronym refer to avian Influenza virus! Should it be in general Influenza A virus

(IAV).

In the section concerning PARP10, it is actually AIV, but in the PARP13 section it was IAV, which  has been changed.

  1. Table 1: PARP10 viruses targeted; there are other viruses should be included as well.

After research in the literature, we could not find other viruses to include in this section.

  1. Line 210: the sentence needs to be rephrased to clarify what was depleted from these cells.

This sentence has been rephrased: « Overexpression of human PARP13 in 293TRex cells reduced production of the HIV-1 Nef protein, which plays an important role in virus replication in vivo, by nearly 13-fold whereas in the same condition nef mRNA expression  decreased by 5-fold, suggesting an inhibition of mRNA translation [19]. Moreover, translation inhibition by PARP13 was not related to mRNA degradation. »

  1. Line 237: the significant of the last sentence is not clear and why the this binding leads to translation inhibition.

This sentence has been rephrased:” PARP12 catalytic activity has been shown to be necessary to downregulate  mRNA translation [61].”

  1. Line 250: a spelling mistake “ arbor ” to “harbor”.

This has been corrected.

  1. Line 255: the word “There” should be “Therefore”.

This has been corrected.

  1. Line 256: the word “bounds” should be “bound” and I suggest to rewrite the sentence.

The sentence has been rewritten:” In addition, decreased  PARP10 expression leads to cell proliferation and promotion of viral replication [66]. »

  1. Line 265: the acronym “K-48” should be defined “Lys 48” or anything else.

This has been corrected.

  1. Line 270: the sentence “…., the immunomodulation properties of PARP1 will not be discussed here [71-73]” should be modified to indicate that these review references discussed the properties of PARP1 and will not discussed in our review.

This has been modified by: ” the immunomodulation properties of PARP1 have been discussed in several reviews and will not be discussed here [71–73]. »

  1. Figure 2:
  2. Define the arrows and stop lines and what do they indicate to.

Arrows and stop line have been defined in the figure directly.

  1. Step number 3 is missing in the figure.

It has been added.

  1. Line 295: this long paragraph should be broken into shorter paragraphs and order the paragraphs to follow figure 2 numbering or re-order the numbering in figure 2 to that in the text.

The paragraph has been totally rewritten and reordered in order to fit with the numbering of the figure 2.

  1. Line 327-329: remove this sentence as it is not related to viruses.

The sentence has been removed.

  1. Line 334: the sentence started with ( In contrast, ….. ) should be re-written using present tense.

The sentence has been modified: « Moreover, it has  recently been shown that ZAP-S binds to and mediates the degradation of several host IFN mRNAs, thereby acting as a negative feedback regulator of the interferon response [37]. »

  1. Line 343 and 344: would be better to keep the term MARylation consistent, choose either mono-ADP-riboylation or MARylation after explaining the term.

The term was explained in the introduction (lines 21-22). In the text, the term MARylation has been kept.

  1. Line 357: the genome of SARS-CoV-2 also has this domain and should be included in to the list of

viruses.

It is now indicated  that both SARS-CoV-1 and -2 own macrodomains:” The genome of several RNA viruses such as coronaviruses ((severe acute respiratory syndrome-related coronavirus (SARS-CoV)-1 and -2, Middle East respiratory syndrome coronavirus (MERS-CoV)),… »

  1. Line 364: “ removing ADP-ribose residues “ not very accurate, macrodomain removes the modification from certain residues but not the residue itself.

This has been corrected:” Macrodomains are capable of recognizing and binding mono- and poly-ADP-ribosylated proteins, subsequently modifying the ADP-ribose residues and downstream signaling or antiviral roles [84–86]. »

  1. Line 369: this study referred to as reference [75] discussed PARP14 only and speculate the same

activity to some other PARPs. Therefore, it would be better to be specific in this.

The other PARPs have been removed:” Interestingly, PARP14 exhibits macrodomains, allowing them to remove ADP-ribose from substrates and exercise regulation or feedback on their PARylation and MARylation activities [79]. »

  1. Line 373: the acronym “AIV” should be “IAV” for Influenza A virus.

This has been corrected.

  1. Line 374: space between “NS1” and “antagonizes”.

This has been corrected.

  1. Line 377: extra word “can”.

This has been corrected.

  1. Line 378: this sentence “ This viral protein …. “ should be re-written because of some grammatical errors.

The sentence has been rewritten:”This viral protein has been shown to be involved in immune response escape [91-92]. PARP13 is responsible for inhibition of EV A71 replication. A recent study suggests that 3C protease cleaves PARP13 at the Gln-369 residue, leading to the loss of this inhibitory activity on the viral replication [93]. »
